# Determination of Safety-Oriented Pavement-Friction Performance Ratings at Network Level Using a Hybrid Clustering Algorithm

**Jieyi Bao** [1] , **Yi Jiang** [1] **and Shuo Li** [2,*]

1    School of Construction Management Technology, Purdue University, West Lafayette, IN 47907, USA; bao59@purdue.edu (J.B.); jiang2@purdue.edu (Y.J.)
2    Division of Research, Indiana Department of Transportation, West Lafayette, IN 47906, USA
*    Correspondence: sli@indot.in.gov

**Abstract:** Pavement friction plays a crucial role in ensuring the safety of road networks. Accurately assessing friction levels is vital for effective pavement maintenance and for the development of management strategies employed by state highway agencies. Traditionally, friction evaluations have been conducted on a case-by-case basis, focusing on specific road sections. However, this approach fails to provide a comprehensive assessment of friction conditions across the entire road network. This paper introduces a hybrid clustering algorithm, namely the combination of density-based spatial clustering of applications with noise (DBSCAN) and Gaussian mixture model (GMM), to perform pavement-friction performance ratings across a statewide road network. A large, safety-oriented dataset is first generated based on the attributes possibly contributing to friction-related crashes. One-, two-, and multi-dimensional clustering analyses are performed to rate pavement friction. After using the Chi-square test, six ratings were identified and validated. These ratings are categorized as (0, 20], (20, 25], (25, 35], (35, 50], (50, 70], and (70, ∞). By effectively capturing the hidden, intricate patterns within the integrated, complex dataset and prioritizing friction-related safety attributes, the hybrid clustering algorithm can produce pavement-friction ratings that align effectively with the current practices of the Indiana Department of Transportation (INDOT) in friction management.

**Keywords:** pavement-friction rating; network level; road-safety attributes; hybrid clustering; density-based spatial clustering of applications with noise (DBSCAN); Gaussian mixture model (GMM); Chi-square test

## 1. Introduction

Pavement friction plays a critical role in ensuring road safety by preventing vehicle tires from sliding or skidding on the road surface. Its primary purpose is to provide adequate traction, especially in wet conditions, between the tires and the pavement [1]. By facilitating traction, pavement friction helps drivers maintain control over their vehicles, significantly reducing the risk of accidents caused by skidding or hydroplaning. This is particularly important at critical locations such as curves, intersections, tunnel entrances, and downhill gradients. In addition, emergency vehicles, buses, heavy trucks, and motorcycles rely heavily on sufficient pavement friction for safe maneuvering. Various factors influence pavement friction, including the texture of the pavement surface, the type of surface material, the properties of the tires, the speed of the vehicle, and the prevailing weather conditions [2,3]. To ensure an acceptable level of pavement friction, regular maintenance and friction treatments are essential. It is also necessary to monitor the performance and conditions of friction, employing measures such as periodic assessments and friction testing [4–6]. These proactive measures contribute to maintaining sufficient pavement friction, thereby enhancing overall road safety.

Pavement-friction performance ratings play a crucial role in assessing the level of friction and the resulting safety provided by a road surface, especially during challenging

weather conditions. These ratings are valuable tools for highway agencies as they assist in identifying areas in need of maintenance and repair. By incorporating these findings into pavement preservation, resurfacing, and overlay programs, optimal performance of the road surface can be ensured. In addition, many countries and roadway or airport authorities have established regulations and standards for pavement-friction performance that must be adhered to in order to ensure safety and prevent accidents [7–10]. Utilizing pavement-friction performance ratings aids in meeting these regulations and standards, guaranteeing compliance and enhancing overall safety.

However, obtaining reliable and objective ratings for pavement friction can be a complex task due to several factors. Firstly, different testing methods and devices can yield varying results, making it difficult to compare data obtained from different testing approaches. Secondly, there is currently no standardized method for assessing pavement-friction performance. Various agencies and organizations adopt different criteria and scales, leading to challenges in comparing results from different sources. Thirdly, the friction of pavement is influenced by its surface texture, especially microtexture, which presents difficulties in accurately and consistently measuring texture. Lastly, despite significant efforts to investigate the impact of pavement friction on vehicle crashes [11–16], a widely accepted correlation has not been established yet. This is primarily due to the fact that vehicle crashes result from the combined effects of human, vehicle, and roadway factors [17]. Each factor encompasses multiple attributes that can individually or collectively influence vehicle crashes, and obtaining information about specific safety attributes may not always be readily available or easily accessible. Consequently, identifying hidden crash patterns within a dataset containing diverse safety attributes using traditional statistical algorithms becomes extremely challenging.

The objective of this paper is to analyze the existing limitations of current methods utilized for assessing pavement performance, specifically in terms of friction. Additionally, the paper aims to develop and implement a safety-focused machine learning algorithm, specifically Gaussian mixture model (GMM)-based clustering, to establish pavement-friction performance ratings, particularly at a network level.

## 2. Literature Review

### 2.1. Threshold-Based Rating Methods

Pavement-performance ratings encompass the evaluation of various factors, including surface distresses, ride quality, structural integrity, and friction, to determine the overall performance or serviceability of the pavement. This assessment aids in prioritizing maintenance and rehabilitation efforts. Presently, several methods are utilized to rate pavement performance, including visual inspection, automated data collection, and non-destructive testing (NDT) [18,19]. Different agencies and organizations may employ variations of these methods or develop customized approaches tailored to their specific requirements and available resources.

Pavement friction, a result of tire-pavement interaction, primarily varies with vehicle speed, tire characteristics, pavement surface texture, and the presence of water. Various methods can be employed to rate pavement friction, depending on the user's specific needs. Friction-threshold rating methods typically involve measuring the friction coefficient, texture, or a combination of both. Friction-coefficient measurements are commonly made using devices such as the locked wheel skid tester (LWST) [20], the British pendulum tester (BPT) [21], or the dynamic friction tester (DFT) [22]. Texture measurements are typically the mean profile depth (MPD) of macrotexture [23] obtained using the sand patch test [24] or noncontact techniques such as the circular track meter (CTM) [25]. The International Friction Index (IFI) combines the friction coefficient and mean profile depth to provide a comprehensive rating of pavement friction, enabling comparisons between different pavements [26].

In the United States, state highway agencies commonly rely on the LWST to obtain friction measurements [2,27]. The threshold-based methods aim to identify friction threshold

values to mitigate vehicle crashes on wet pavement. This simplifies the rating process by conducting field tests to measure friction and comparing the results against the threshold value. If the measured friction falls below the threshold value, appropriate actions may be required to restore adequate friction levels. Table 1 provides an overview of the friction threshold values recommended by different researchers. The table illustrates significant variations in the threshold values between researchers and highway agencies. These variations can be attributed to two primary factors. Firstly, some agencies utilize standard rib tires [28], while others use standard smooth tires [29]. Friction measurements with rib tires are considerably higher than those obtained with smooth tires. Secondly, researchers employ diverse datasets and consider various factors, resulting in substantial discrepancies in the recommended threshold values.

**Table 1.** Friction threshold values for remedial actions.

| Source | Test Condition | Threshold Value |
|---|---|---|
| Kummer and Meyer [30] | Rib tire, 40 mph | 37 |
| Henry [2] | Rib or smooth tire, 40 mph | 30~45 |
| Noyce et al. [31] | Rib tire, 40 mph | 35 |
| Kuttesch [32] | Smooth tire, 40 mph | 25~30 |
| Li et al. [33] | Smooth tire, 40 mph | 20 |
| Zhao et al. [15] | Smooth tire, 40 mph | 20 |

The above rating method and the like offer two advantages. Firstly, they provide a straightforward and measurable evaluation of pavement friction, based on predetermined engineering thresholds. Secondly, the threshold values establish a standardized criterion for assessing and comparing pavement friction, which ensures consistency across different sections of pavement and is essential for crash prevention, especially in adverse weather conditions. However, the arbitrary threshold values lack a robust scientific foundation and exhibit inconsistencies among different highway agencies. By adopting an arbitrary friction threshold, essential contextual factors such as vehicle speed, traffic volume, road geometry, weather conditions, and related costs may not be accurately evaluated. Moreover, pavement friction is a dynamic property that constantly changes due to weather, traffic, and various other factors. An arbitrary threshold may fail to account for these variations or provide a mechanism to adjust the threshold based on evolving conditions. Consequently, threshold-based rating methods fall short in delivering adequate warning or transition time to implement preventive measures, leading to missed opportunities for timely maintenance.

*2.2. Multilevel-Based Rating Methods*

Recently, a novel trend has surfaced in the evaluation of pavement friction, which involves the utilization of supervised learning techniques to assess pavement friction across multiple levels. Noteworthy contributions in this domain include the research endeavors of Zhan et al. [34] and Zhao et al. [35]. The former introduced an innovative approach employing a deep residual network (ResNets) to predict pavement friction using surface texture. On the other hand, the latter demonstrated the application of extreme gradient boosting (XGBoost) to establish a correlation between friction and safety. Given the subject matter of this paper, this section primarily provides a brief introduction to the work of Zhao et al. In their research work, the XGBoost model was utilized to classify crash severity, identify the contributing factors through the model outputs, and quantify the relationships between friction and crash severity. Their work yielded five pavement friction classes based on the friction numbers (FNs): FNS < 20, FNS $\in$ (20, 25), FNS $\in$ (25, 38), FNS $\in$ (38, 70), and FNS > 70.

Evidently, the above multilevel classification method can offer a more comprehensive, informed, and systematic approach to assessing and managing pavement friction and aid in decision-making, planning, budgeting, and performance monitoring. Nevertheless, the classifications can often become problematic. An example is that within FNS $\in$ (20, 25), a

significant number of observations exhibit a lower probability of fatal or injury crashes. This is likely because there are several crucial safety attributes, such as the vehicle speed at the time of crashing and the pavement friction at the crash location that cannot be accessible or accurately determined. Although the analyzed datasets included friction, vehicle, and crash attributes, no class labels or target values were assigned to them. Employing supervised learning techniques such as the XGBoost model to ascertain the collective impact generated by these attributes is exceedingly challenging.

## 3. Data and Integration

### 3.1. Datasets

Two types of datasets, pavement friction and vehicle crashes, are used in this paper. The pavement-friction data was obtained through the annual pavement-friction inventory test program of the Indiana Department of Transportation (INDOT) [3,33]. This program comprises four main components: in-house system calibration, field testing, data processing, and reporting. Field friction testing is carried out at one-mile intervals using the LWST on all interstate highways in both directions each year, while on other routes, such as US highways and state roads, testing is conducted in one direction every three years. Friction is measured in the left wheel track of the driving lane using a standard smooth tire, at speeds of 48 km/h (30 mph), 64 km/h (40 mph), or 80 km/h (50 mph). The calculation method is shown below [20]:

$$SN = F/W \times 100 \tag{1}$$

where *SN* = skid number that is used interchangeably with friction number (*FN*) in this paper, *F* = horizontal force applied to the test tire at the tire-pavement contact patch, lbf (or N), and *W* = dynamic vertical force on the test wheel, lbf (or N).

The obtained friction dataset consists of attributes, such as geographic region (district and county), road details (name, direction, test lane, and type of road surface [asphalt, concrete, or bridge deck]), test conditions (speed and temperature), and test results (friction number at the actual test speed and friction number converted to the standard test speed of 40 mph). Additionally, the dataset includes test location indicated by reference post (RP) and global positioning system (GPS) coordinates. Notably, the friction dataset spans three consecutive years: 2017, 2018, and 2019, resulting in a substantial collection of 25,458 data points. This approach ensures comprehensive coverage of the entire road network under the jurisdiction of INDOT.

The vehicle crash dataset was obtained through the Automated Reporting Information Exchange System (ARIES) [36] of the Indiana State Police (ISP). ARIES serves as a central repository for capturing, organizing, and reporting information related to vehicle crashes that occur within the state of Indiana. In line with the collected pavement-friction dataset, the vehicle-crash dataset was also generated for the years 2017, 2018, and 2019, comprising fifty-nine attributes, providing detailed information about each crash event. The available details include the date, time, location, road conditions (geometrics, surface type, median type, and junction type), weather conditions, light conditions, traffic control, types of vehicles involved, primary factors contributing to the crash, manner of collision, and collision outcomes. The combination of crash data from these three years yields a total of 200,145 crash events.

### 3.2. Integration

To ensure the availability of a comprehensive and robust dataset suitable for safety-oriented friction analysis and evaluation, the integration of pavement-friction data and vehicle-crash data was conducted in four steps as follows:

- Data reorganization: Using the year and road type (i.e., interstate highways, US highways, and state roads) obtained from both datasets, the crash and friction data were grouped in pairs, resulting in the formation of nine groups of sub-datasets.

- Spatial integration: Using geographic information systems (GIS) coordinates or reference posts, the distances between each crash event location and all friction-test locations were calculated. Each crash event was then linked to the friction measurement with the shortest distance, which is commonly 1 mile or less, considering the interval of friction testing by INDOT.
- Data merging: All data points generated in the spatial-integration step were merged based on the road name.
- Safety-oriented filtering: A meticulous filtering approach was implemented to eliminate crash events caused by factors unrelated to friction. These factors included vehicle malfunctions, driver usage of cellphones or telematics, and driver illness.

The data-integration preprocessing described above was implemented using the Python 3.9 programming language. The integrated dataset contained 29,136 data entries, each characterized by 5 safety-related variables (or attributes), namely friction number, crash-severity level, surface condition, road geometrics, and pavement-surface type. These variables are presented in Table 2. Apart from the friction number, all other variables are categorical in nature. Specifically, the "crash-severity level" variable encompasses two categories: property damage only (PDO) and injury or fatality. The "surface condition" variable comprises four groups: dry, wet/water, ice, and snow/others (slush/muddy/loose materials on road surface). Road geometrics includes three types: grade, level, and hillcrest. The "surface material" variable consists of three types: asphalt, concrete, and gravel. To ensure consistency, all data in the dataset underwent a standardization process, scaling the values to a uniform magnitude.

**Table 2.** Variable descriptions.

| Variables | Description | Categories | | | |
|---|---|---|---|---|---|
| FNS | Friction numbers | NA | | | |
| Crash-Severity Level | Property damage only (PDO), and injury or fatality. | 0 = PDO (87.94%) | 1 = Injury or Fatality (12.06%) | | |
| Surface Condition | Affected by weather when crashes happened. | 1 = Dry (71.57%) | 2 = Wet/Water (15.91%) | 3 = Ice (5.99%) | 4 = Snow/Others (6.52%) |
| Road Geometrics | Grade, level, and hillcrest. | 1 = Grade (17.18%) | 2 = Level (78.44%) | 3 = Hillcrest (4.38%) | |
| Surface Material | Asphalt, concrete, and gravel | 1 = Asphalt (73.42%) | 2 = Concrete (26.53%) | 3 = Gravel (0.5%) | |

## 4. Methodology

### 4.1. Density-Based Spatial Clustering of Applications with Noise

The density-based spatial clustering of applications with noise (DBSCAN) algorithm is a widely employed density-based clustering technique known for its ability to identify clusters of various shapes within a dataset, while effectively handling noise and outliers [37,38]. Unlike other clustering algorithms, DBSCAN does not require a predefined number of clusters. Clusters are identified based on the density characteristics of the data. This makes DBSCAN particularly advantageous for clustering tasks where the number of clusters is unknown or variable. DBSCAN utilizes two critical hyperparameters: epsilon ($\varepsilon$), i.e., the radius for defining neighboring points within the same cluster, and the minimum number of points (minPts) to set the threshold for forming dense regions or clusters. The appropriate values for $\varepsilon$ and minPts need to be predetermined. Typically, $\varepsilon$ can be set to the average distance between points or the distance at which the k-nearest neighbors graph achieves connectivity. The minPts can be established as the dimensionality of the data

plus one. To determine the optimal hyperparameters for DBSCAN, the silhouette score is commonly employed and calculated as follows:

$$s = \frac{b - a}{\max(a, b)} \tag{2}$$

where $s$ = silhouette score; $a$ = the average distance between an object and all other objects in the same cluster; and $b$ = the average distance between an object and all other objects in the next nearest cluster.

The resulting silhouette score, ranging from $-1$ to 1, serves as a performance metric for evaluating the quality of clustering. A silhouette score approaching -1 signifies that the clustering outcome is incorrect or poorly separated. Conversely, a silhouette score nearing 1 suggests a higher clustering density and a more accurate clustering result. A silhouette score close to 0 implies that the clustering is sketchy or the data points are close to decision boundaries between clusters. The silhouette score aids in quantitatively assessing the efficacy of the clustering algorithm and determining the appropriateness of the chosen hyperparameters. Once $\varepsilon$ and minPts have been determined, the clustering process is performed as illustrated elsewhere [37,38].

### 4.2. Gaussian Mixture Model

The Gaussian mixture model (GMM) is a probabilistic model that postulates the data as originating from a mixture of multiple Gaussian distributions [38], wherein the parameters of these distributions are unknown. The calculation of a multivariate Gaussian distribution can be expressed as follows:

$$N(\boldsymbol{x}|\boldsymbol{\mu}, \boldsymbol{\Sigma}) = \frac{1}{\sqrt{(2\pi)^d|\boldsymbol{\Sigma}|}} \exp\left\{-\frac{1}{2}(\boldsymbol{x} - \boldsymbol{\mu})^T\boldsymbol{\Sigma}^{-1}(\boldsymbol{x} - \boldsymbol{\mu})\right\} \tag{3}$$

where $\boldsymbol{x}$ = a d-dimensional random vector, and $\boldsymbol{x} \sim N(\boldsymbol{\mu}, \boldsymbol{\Sigma})$; $\boldsymbol{\mu}$ = a d-dimensional mean vector; and $\boldsymbol{\Sigma}$ = a d $\times$ d covariance matrix.

GMM requires determining the optimal number of clusters in advance. To ascertain the optimal number of clusters for the GMM, it is beneficial to choose the model that minimizes a theoretical information criterion. Two commonly employed metrics for determining the optimum cluster number in the GMM are the Bayesian Information Criterion (*BIC*) and the Akaike Information Criterion (*AIC*) expressed as follows [39,40]:

$$BIC = \log(n)p - 2\log(\hat{L}) \tag{4}$$

$$AIC = 2p - 2\log(\hat{L}) \tag{5}$$

where $n$ = the number of points; $p$ = the number of parameters learned by the model; and $\hat{L}$ = the maximized value of the likelihood function of the model. Notice that although *BIC* and *AIC* often yield similar results, there can be instances where they diverge. In such cases, *BIC* tends to prefer simpler models, while *AIC* may select a model that is better tailored to the specific characteristics of the data.

Maximum likelihood estimation is a widely adopted method for deriving parameters in statistical models. However, in scenarios where the underlying distribution generating the dataset is unknown, utilization of the maximum likelihood principle for parameter estimation becomes challenging. In such cases, the Expectation-Maximization (EM) algorithm is employed. The EM algorithm consists of two main steps: the exception step and the maximization step. The expectation step involves estimating the missing or unobserved variables. In the maximization step, the new set of estimated parameters is determined to maximize the expected likelihood, which optimizes the parameters based on the completed data, incorporating the information from the expectation step. The expectation step and maximization step iteratively alternate until a stopping criterion is met. This criterion

can be defined as either the parameters reaching a state of convergence where they no longer change significantly or the change in parameters falling below a predetermined threshold. This iterative process allows for the refinement of parameter estimates, gradually improving the accuracy of the model [38].

### 4.3. DBSCAN-GMM Algorithm

The DBSCAN-GMM algorithm is a hybrid clustering approach that combines the strengths of both DBSCAN and GMM, aiming to improve clustering performance and mitigate their respective limitations. The DBSCAN-GMM algorithm is more suitable for datasets containing clusters with diverse density distributions and more effective for identifying noisy points in the presence of overlapping clusters or data with complex distributions. It has enhanced the ability to distinguish genuine clusters from noise and reduced sensitivity, leading to more accurate and robust cluster assignments, reliable cluster assignments, and improved clustering accuracy.

The hybrid algorithm first utilizes the DBSCAN algorithm to identify the core points and border points within the dataset. Then, the GMM is applied to effectively model the underlying distribution within each identified cluster. This approach enables effectively capturing intricate patterns hidden in the data, including the presence of clusters exhibiting multiple modes. By integrating the strengths of DBSCAN and GMM, the hybrid algorithm is capable of handling complex data structures and accurately representing the underlying distribution within each cluster.

### 4.4. Chi-Square Test

In the context of this paper, the Chi-square test is employed as a tool to determine the final pavement-friction performance ratings. The rationale behind the utilization of the Chi-square test can be attributed to several key factors: firstly, the Chi-square test is well-suited for analyzing both categorical and continuous data, which is congruent with the dataset employed in this investigation; secondly, the Chi-square test tends to exhibit enhanced accuracy when applied to larger datasets, thereby bolstering the reliability of the obtained results; lastly, the Chi-square test assumes the independence of observations and the absence of extremely low expected frequencies in each category, aligning with the specific characteristics of the data under examination [41]. Specifically, it is used to investigate the association or independence between variables within each friction performance rating. The properties of the dataset utilized in this study meet the requirements of the Chi-square test, specifically in terms of independence and the large sample size, which ensure the accuracy of the test [41]. The independence of observations assumes that the data points are not influenced by each other, allowing for unbiased analysis. Additionally, a sufficiently large sample size is crucial for the Chi-square test to yield reliable results, as it increases the validity and robustness of the statistical analysis. By analyzing the association between these variables, valuable insights can be gained regarding the impact of different factors on pavement-friction performance.

To conduct a Chi-square test, the initial step involves stating the null hypothesis ($H_0$) and the alternative hypothesis ($H_a$) based on the specific research question. Following this, a contingency table is created to summarize the observed frequencies of the categorical variables under investigation. This table provides a structured representation of the relationship between the variables. The Chi-square statistic is then calculated by comparing the observed frequencies with the expected frequencies. The formula to calculate the Chi-square test statistic is as follows:

$$\chi^2 = \sum \frac{(O - E)^2}{E} \tag{6}$$

where $\chi^2$ = the Chi-square statistic; $O$ = the observed frequency in each cell of the contingency table, i.e., the actual count observed in the sample data for each combination of categories in the variables being analyzed; and $E$ = the expected frequency in each cell

under the assumption of independence, i.e., the frequency expected if the variables were independent, calculated based on the marginal totals and assuming independence.

A larger Chi-square value indicates a stronger discrepancy between the observed and expected frequencies, suggesting a more significant association between variables. When performing a Chi-square test, the calculated Chi-square test statistic is compared to the critical value from the Chi-square distribution, considering the degrees of freedom. The degree of freedom for the Chi-square test is determined by the dimensions of the contingency table. If the number of rows is denoted by r and the number of columns by c, then the degree of freedom is calculated as:

$$df = (r - 1) \times (c - 1) \tag{7}$$

The Chi-square distribution and the degree of freedom are employed for computing the *p*-value associated with the Chi-square statistic. If the *p*-value is below a predetermined significance level (usually 0.05), the null hypothesis of independence is rejected. This provides evidence of an association between the variables being examined.

## 5. Results and Analysis

### 5.1. One-Dimensional Clustering Analysis

The one-dimensional DBSCAN-GMM algorithm was first developed to investigate the attributes of friction numbers (FNS). Figure 1 portrays the kernel density estimation distribution of FNS. This visual combines a histogram-like depiction of the data, showcasing the frequency distribution, with a smooth curve that represents the estimated probability density function. The abscissa of the graph corresponds to the friction numbers acquired at a velocity of 40 miles per hour, while the ordinate represents the density values associated with the respective friction number samples. The quantity of columns depicted in Figure 1 is established utilizing the Freedman-Diaconis rule, an algorithmic procedure. The figure displays a total of 85 columns, wherein the height of each column conveys the proportional representation of the friction number samples falling within a specific range or interval of friction number values. The graphical representation reveals a left-skewed distribution pattern. The 25th, 50th, and 75th percentiles correspond to values of 27.6, 38.2, and 48.8, respectively. Notably, approximately 95% of the friction numbers fall within the range of 15.5 to 71.8.

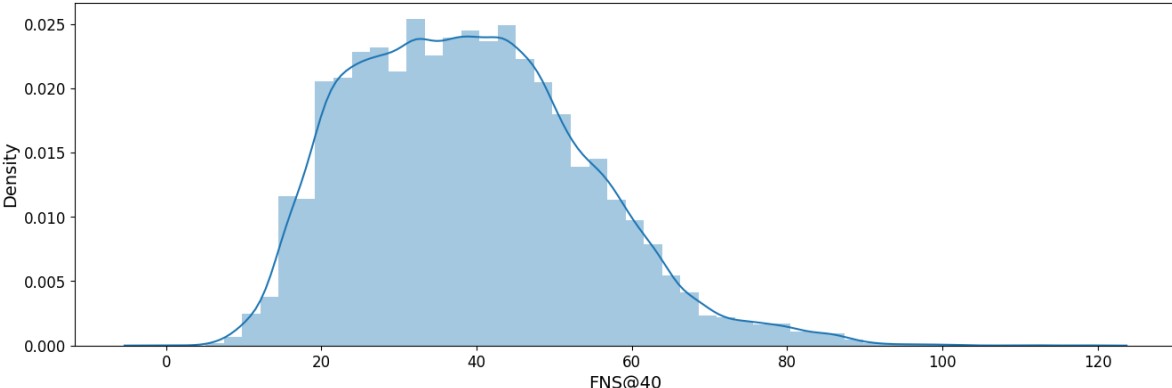

**Figure 1.** Kernel density estimate of friction numbers. Note: FNS@40 denotes the friction numbers measured at 40 mph.

To implement the DBSCAN-GMM algorithm, the hyperparameter $\varepsilon$ and minPts were fine-tuned to values of 0.1 and 5, respectively. The optimum number of clusters in the GMM was determined by evaluating both *BIC* and *AIC* scores as shown in Figure 2. Based on the *AIC* and *BIC* scores, the optimum number of clusters for the 1-dimensional DBSCAN-GMM model is determined to be five. Additionally, Figure 3 displays the cumulative frequency distribution (CFD) of the clustered FNS. The results indicate that the FNS are initially

divided into five distinct groups, as presented in Table 3. However, to further validate the effectiveness of the clustering results, additional variables need to be explored.

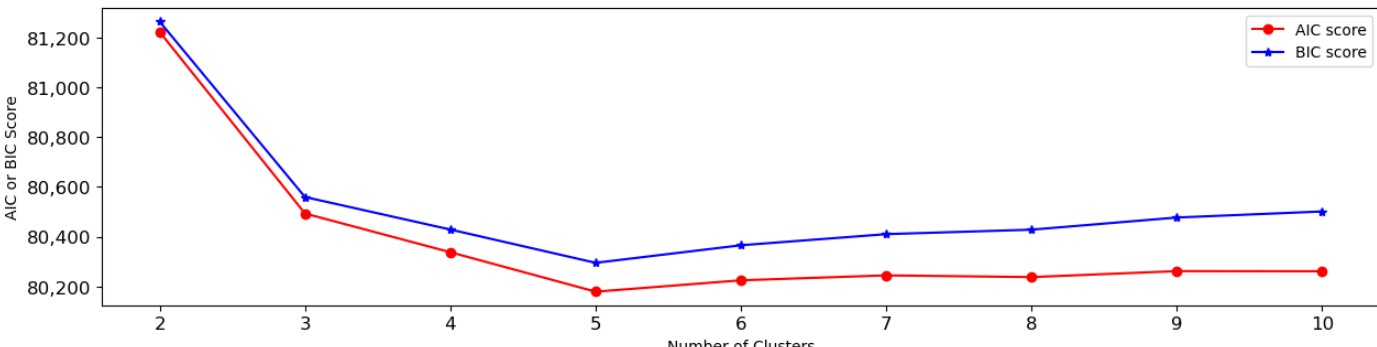

**Figure 2.** *AIC* and *BIC* score for 1-D DBSCAN-GMM model.

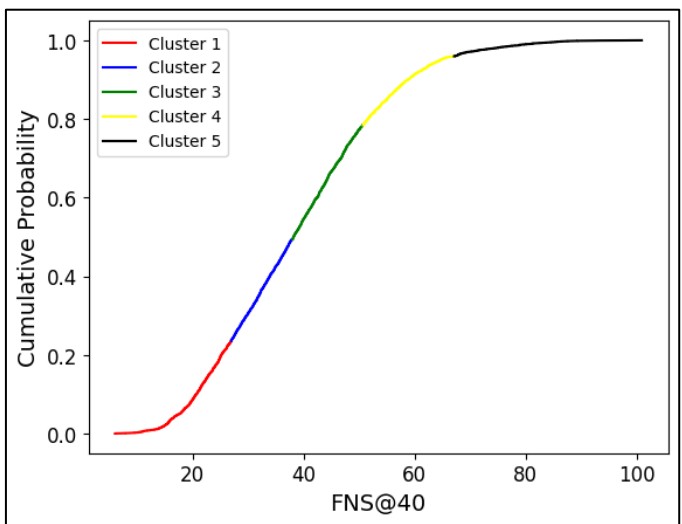

**Figure 3.** CFD of clustered friction numbers generated by 1-D DBSCAN-GMM model. Note: FNS@40 denotes the friction numbers measured at 40 mph.

**Table 3.** Friction performance ratings based on 1-D DBSCAN-GMM model.

| No. | Friction Number (FN) |
| --- | --- |
| 1 | $0 < FN \leq 25$ |
| 2 | $25 < FN \leq 35$ |
| 3 | $35 < FN \leq 50$ |
| 4 | $50 < FN \leq 70$ |
| 5 | $FN > 70$ |

### 5.2. Two-Dimensional Clustering Analysis

Two-dimensional clustering was performed by including the influence of crashes. Notice that 88% of the crashes in the dataset were property-damage-only accidents. The hyperparameters $\varepsilon$ and minPts were fine-tuned, resulting in values of one and three, respectively. Figure 4 presents the *AIC* and *BIC* scores obtained from the 2-D DBSCAN-GMM model. The optimum number of clusters determined from these scores is eight. Evidently, there are overlaps among clusters obtained from 1-D and 2-D DBSCAN-GMM models. The eight clusters can be divided into two groups. The first one solely consists of PDO crashes, while the second group exclusively comprises injury or fatal crashes. Among these eight clusters, five clusters are formed by friction numbers associated with PDO

crashes, while the remaining three clusters are generated by friction numbers associated with injury or fatal crashes.

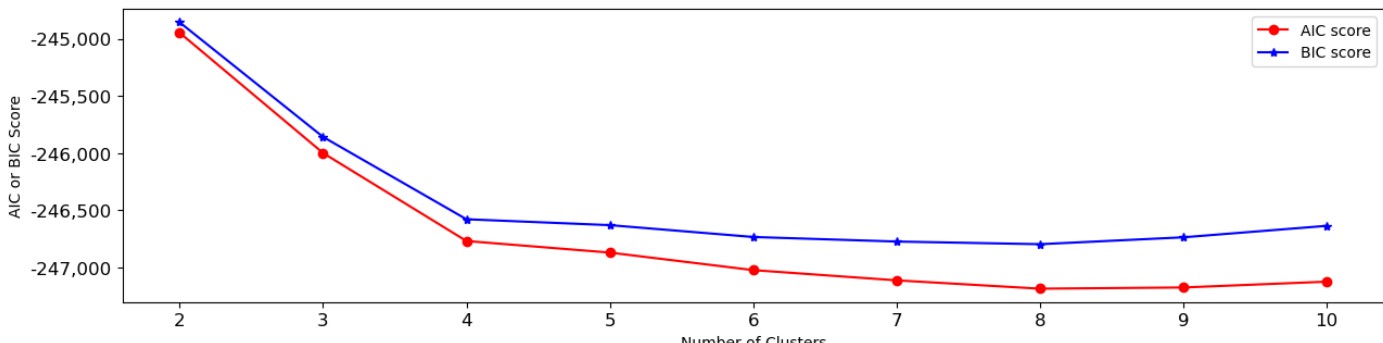

**Figure 4.** *AIC* and *BIC* score for 2-D DBSCAN-GMM model.

Figure 5 presents the cumulative frequency distribution (CFD) of the clustered friction numbers obtained using the 2-D DBSCAN-GMM model. Figure 5a displays the CFD of friction numbers associated with property-damage-only crashes, while Figure 5b represents the CFD of friction numbers linked to injury or fatal crashes. The friction-number range within each cluster of PDO crashes remains consistent with the ranges generated in the 1-D model. Additionally, the friction-number range of cluster 6 associated with PDO crashes is almost equal to the combined friction-number ranges of clusters 1 and 2. Similarly, the friction-number range of cluster 7 associated with PDO crashes is nearly equal to the combined friction-number ranges of clusters 3 and 4. Furthermore, the friction-number range of cluster 8 associated with PDO crashes closely resembles the friction-number range of cluster 5. Therefore, the pavement-friction ratings generated by the 2-D DBSCAN-GMM model are basically consistent with the results generated by the 1-D model as shown in Table 3.

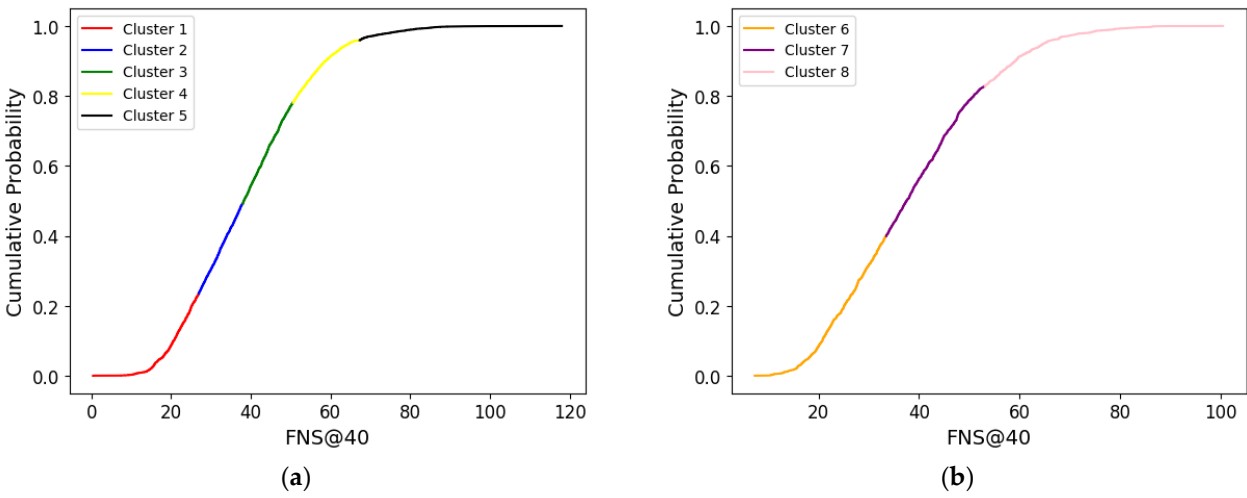

**Figure 5.** CFD of clustered friction numbers generated by 2-D DBSCAN-GMM model. (**a**) CFD of friction number with property-damage-only crashes, and (**b**) CFD of friction number with injury or fatal crashes.

### 5.3. Multi-Dimensional Clustering Analysis

In addition to friction numbers and crash-severity level, three other variables, namely surface condition, road geometrics, and surface material, were further included in the clustering analysis. Through fine-tuning the hyperparameters, the optimum values of $\varepsilon$ and minPts were determined as 0.5 and 5, respectively. As the number of variables increased, the optimal number of clusters also increased accordingly. Hence, the optimal

cluster number for the multi-dimensional model was identified as 48. Figure 6 illustrates the *AIC* and *BIC* scores for varying cluster numbers. Similar to the 2-D DBSCAN-GMM model, there are observed overlaps among clusters in the friction numbers.

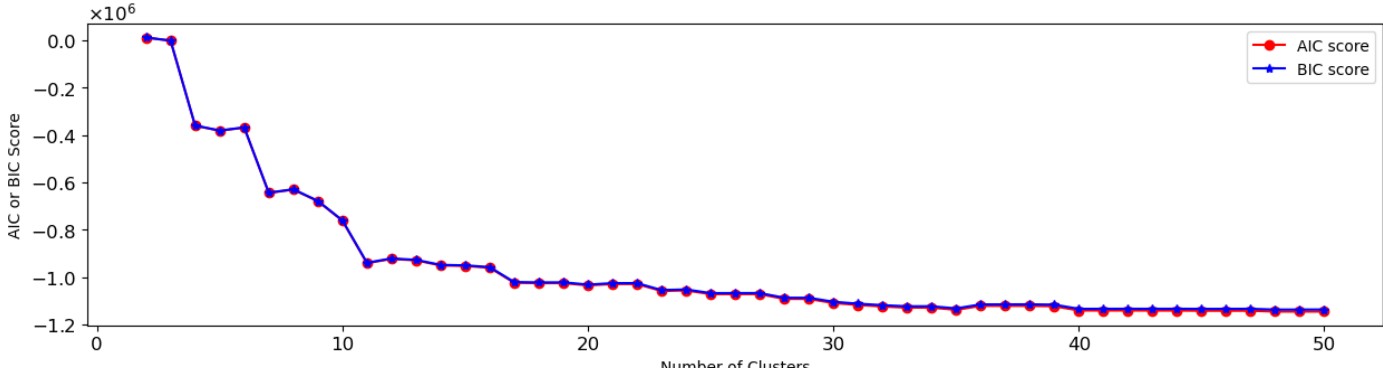

**Figure 6.** AIC and BIC score for Multi-Dimensional DBSCAN-GMM model.

These 48 clusters were further grouped into 30 distinct categories based on the combination of crash severity, surface condition, road geometrics, and surface material. Out of these 30 groups, 7 groups consist of 2 clusters or more, while the remaining 23 groups comprise only 1 cluster each. This is because the friction-number ranges within these clusters cover the entire range of friction numbers. Figure 7 depicts the CFD of the seven groups that contain two clusters or more. In group 1, characterized by the combination of categorical variables [PDO, Dry, Level, Asphalt], the friction numbers are divided into six clusters, as shown in Figure 7a. The clustering results of the multiple-dimensional model are largely consistent with those of the 1-D and 2-D models, except for the friction-number range of (35, 50]. While the 1-D and 2-D DBSCAN-GMM models assign these friction numbers to a single cluster, the multiple-dimensional model divided them into two separate clusters, namely (35, 45] and (45, 50]. In group 2, distinguished by the categorical variable combination [PDO, Dry, Level, Concrete], the clustering results of the multiple-dimensional model combine the friction-number ranges of (25, 35] and (35, 50], which were divided into two separate clusters by the 1-D and 2-D DBSCAN-GMM models, as shown in Figure 7b. For other groups with three clusters or two clusters, the demarcation points occur around 35 and 70. Notably, the clustering results for group 3 and group 5, as shown in Figure 7c,e, exhibit slight differences compared to other groups, as there is an overlapping segment between clusters, specifically in the friction-number range between 0 and 20. Based on the above results, the friction performance ratings generated by multiple-dimensional clusters are summarized and listed in Table 4.

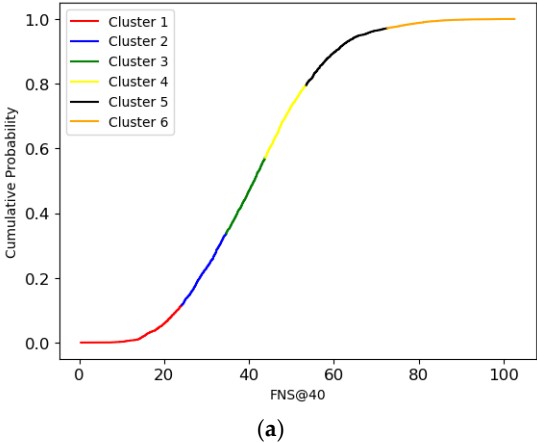

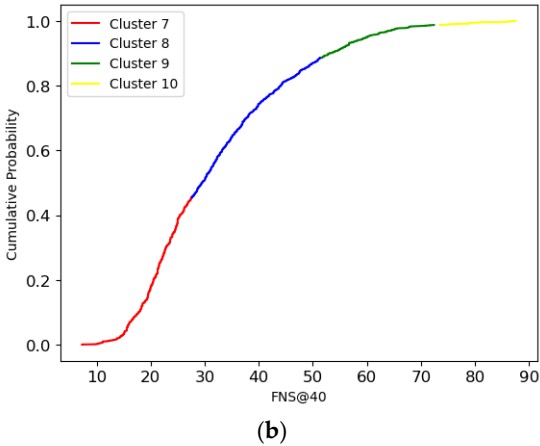

(a)

(b)

**Figure 7.** *Cont.*

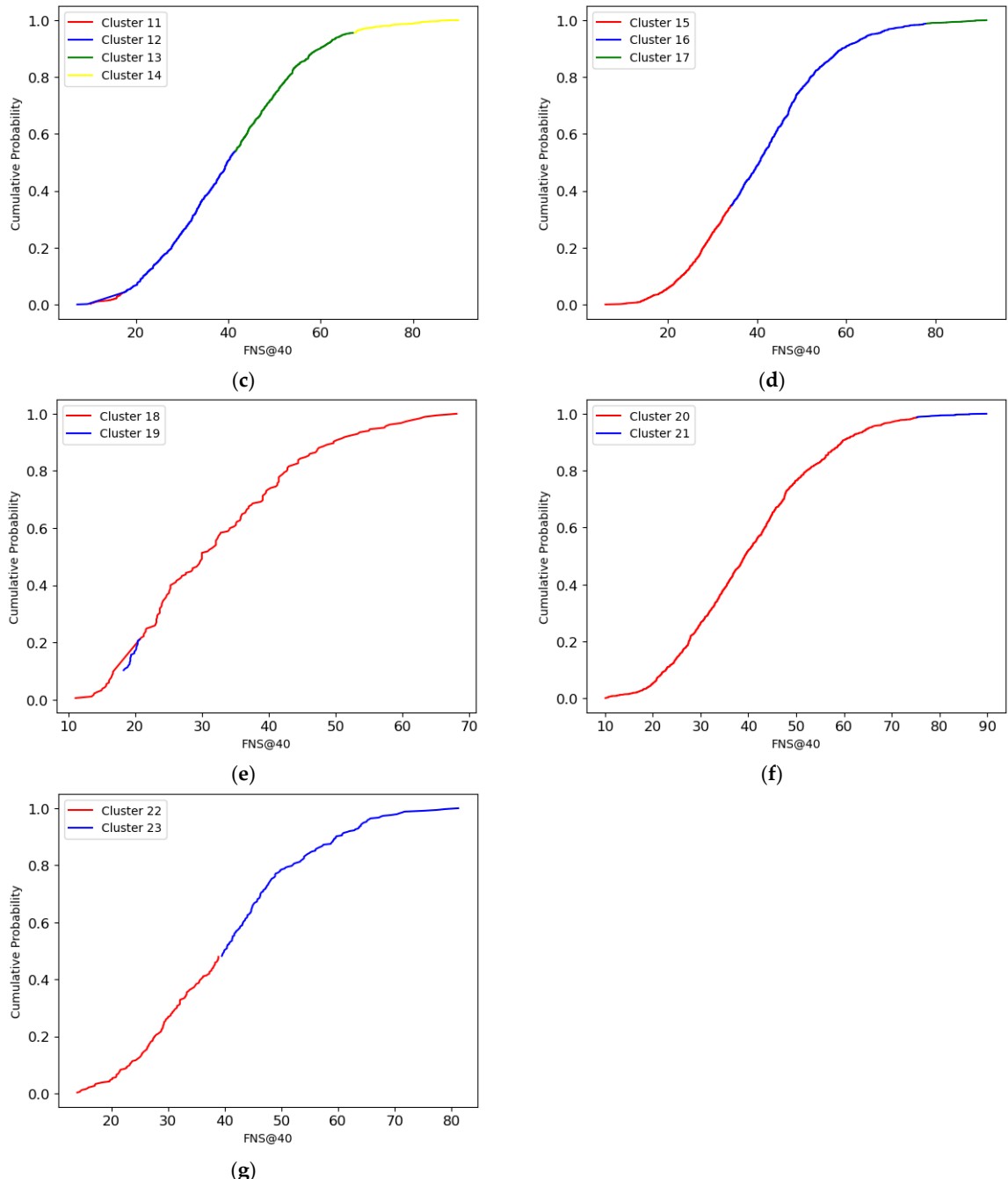

**Figure 7.** CFD of clustered friction numbers generated by the multiple-dimensional DBSCAN-GMM model. (**a**) CFD of group 1; (**b**) CFD of group 2; (**c**) CFD of group 3; (**d**) CFD of group 4; (**e**) CFD of group 5; (**f**) CFD of group 6; and (**g**) CFD of group 7.

**Table 4.** Friction performance ratings based on multiple-dimensional DBSCAN-GMM model.

| No. | Friction Number (*FN*) |
|---|---|
| 1 | $0 < FN \leq 20$ |
| 2 | $20 < FN \leq 25$ |
| 3 | $25 < FN \leq 35$ |
| 4 | $35 < FN \leq 45$ |
| 5 | $45 < FN \leq 50$ |
| 6 | $50 < FN \leq 70$ |
| 7 | $FN > 70$ |

*5.4. Chi-Square Test Analysis*

A Chi-square test was employed to further determine the most appropriate pavement-friction performance ratings. The purpose of the Chi-square test is to assess the independence between the crash-severity level and the three other variables, namely surface condition, road geometrics, and surface material. The null ($H_0$) and alternative ($H_1$) hypotheses for the Chi-square test are as follows:

$H_0$ : *Severity level and $v_i$ are independent.*
$H_1$ : *Severity level and $v_i$ are dependent.*

where $v_i$ = one of the three variables; and $i$ = surface condition, road geometrics, and surface material, respectively.

The significance level is established at 0.05, which serves as a threshold for decision-making in the Chi-square test. This significant level helps determine the statistical significance of the relationship between the severity level and the variables under investigation. If the calculated *p*-value is below 0.05, it indicates sufficient evidence to reject the null hypothesis of independence. As a result, we can infer that the severity level is dependent on the variable "$v_i$". The rejection of the null hypothesis suggests a statistically significant association or relationship between the severity level and the corresponding variable. If the relationship between the crash-severity level and the other variables in two consecutive clusters is found to be the same, it is permissible to combine these two clusters. However, if the relationship differs between the two clusters, they cannot be merged. This criterion ensures that clusters with similar patterns and associations between the severity level and the three variables are grouped together, while clusters with distinct patterns are kept separate. The decision to combine or separate clusters is based on the consistency or inconsistency of the relationships observed between the crash-severity level and the other three variables across consecutive clusters.

Table 5 presents the outcomes obtained from the Chi-square tests. The Chi-square test results indicate that, within the friction range of (0, 20], there is a significant relationship between the severity level and the surface condition. Similarly, for the friction ranges of (25, 35] and (50, 70], the crash-severity level exhibits an association with the surface material. In contrast, across all clusters, the severity level appears to be independent of the surface condition, road geometrics, and surface materials within the range of (35, 45] and (45, 50]. An additional Chi-square test was employed to further examine the relationship between the severity level and the remaining three variables within the range of (35, 50]. The resulting p-values of 0.29, 0.41, and 0.69 all surpass the 0.05 significance threshold, leading to the acceptance of all hypotheses. These findings are consistent with the Chi-square test outcomes of (35, 45] and (45,50], indicating that it is justifiable to combine these two contiguous ranges.

The analysis conducted using the DBSCAN-GMM models and the Chi-square tests has led to the identification of six distinct friction performance ratings, which are summarized in Table 6. Figure 8 displays the CFD figures and relevant statistical values of the entire dataset based on the friction performance ratings in Table 6. For the entire dataset, the mean and median values of the FNS are 39.19 and 38.2, respectively. These values fall within the interval of 35–50, indicating consistent central tendency within this range. Similarly, the 25th percentile of the FNS distribution for each road type falls in the range of 25-35. Additionally, the 75th percentile for each road type has FNS values of around 50. The mean and median values of the FNS for different road types, including interstate highways, state roads, and US highways, exhibit a similar pattern to the overall dataset. This consistency emphasizes the statistical validity of the groupings within the ranges of 25–35, 35–50, and 50–70.

**Table 5.** Chi-square test results.

| FNS | Variable 1 | Variable 2 | $\chi^2$ | df | *p*-Value | Accept H$_0$ or Not |
|---|---|---|---|---|---|---|
| | | Surface Condition | 20.0990 | 3 | 0.0002 | Reject |
| $0 < FN \leq 20$ | Severity Level | Road Geometric | 4.7952 | 2 | 0.0909 | Accept |
| | | Surface Material | 0.6405 | 2 | 0.7260 | Accept |
| | | Surface Condition | 5.4039 | 3 | 0.1445 | Accept |
| $20 < FN \leq 25$ | Severity Level | Road Geometric | 0.7559 | 2 | 0.6853 | Accept |
| | | Surface Material | 4.3729 | 2 | 0.1123 | Accept |
| | | Surface Condition | 2.8680 | 3 | 0.4124 | Accept |
| $25 < FN \leq 35$ | Severity Level | Road Geometric | 1.6748 | 2 | 0.4328 | Accept |
| | | Surface Material | 7.7221 | 2 | 0.0210 | Reject |
| | | Surface Condition | 5.6471 | 3 | 0.1301 | Accept |
| $35 < FN \leq 45$ | Severity Level | Road Geometric | 0.2079 | 2 | 0.9013 | Accept |
| | | Surface Material | 0.5696 | 2 | 0.7522 | Accept |
| | | Surface Condition | 1.9873 | 3 | 0.5751 | Accept |
| $45 < FN \leq 50$ | Severity Level | Road Geometric | 3.1309 | 2 | 0.2090 | Accept |
| | | Surface Material | 0.1312 | 2 | 0.9365 | Accept |
| | | Surface Condition | 3.9647 | 3 | 0.2653 | Accept |
| $50 < FN \leq 70$ | Severity Level | Road Geometric | 0.0896 | 2 | 0.9562 | Accept |
| | | Surface Material | 6.7126 | 2 | 0.0349 | Reject |
| | | Surface Condition | 3.0918 | 3 | 0.3777 | Accept |
| $FN > 70$ | Severity Level | Road Geometric | 1.0527 | 2 | 0.5908 | Accept |
| | | Surface Material | 1.3914 | 1 | 0.2382 | Accept |

Figure 9 visualizes the characteristics of crash-severity level in relation to the other variables for each friction-performance rating using 100% stacked columns. The initial exploration of associations between severity levels and surface conditions within the friction range of (0, 20], as well as between severity levels and surface materials within the friction intervals of (25, 35] and (50, 70], was conducted based on the results obtained from the Chi-square test. Consistent with previous studies [42], injury or fatal crashes are more likely to occur when the road surface is wet or covered in ice, snow, or other loose coverings as shown in Figure 9a. When FNS is less than 20, the severity of the collision's consequences escalates due to the reduced friction properties of the road surface, particularly when encountering icy and snowy conditions. Similar situations can also be observed when the FNS falls within the range of 20 and 25, where there is a higher likelihood of injury or fatal crashes occurring on the road surface with water, ice, snow, or other loose coverings. This coincides with previous research [15]. The proportion of crashes on icy surfaces is substantially higher, with a statistical result showing that more than 25% of crashes on icy surfaces lead to injury or fatality, compared to around 12% in other clusters, as shown in Figure 9b. Additionally, Figure 9c,d demonstrate the relationship between friction and the occurrence of crashes on different road surfaces. As friction increases, the proportion of crashes on concrete roads gradually decreases. However, at high friction values, the proportion of injured or fatal crashes on concrete roads begins to increase gradually, while the proportion on asphalt roads decreases gradually. In the FNS interval of (25, 35], 40% of gravel surfaces are concentrated within this group. Approximately 50% of accidents transpiring on gravel roads result in either injuries or fatalities. In the range of (50, 70], concrete roads exhibit a slightly higher susceptibility to injury and fatal crashes, whereas asphalt roads display a slightly higher likelihood of the occurrence of PDO crashes.

**Table 6.** Final friction performance ratings.

| No. | Friction Number (FN) |
|-----|----------------------|
| 1 | $0 < FN \leq 20$ |
| 2 | $20 < FN \leq 25$ |
| 3 | $25 < FN \leq 35$ |
| 4 | $35 < FN \leq 50$ |
| 5 | $50 < FN \leq 70$ |
| 6 | $FN > 70$ |

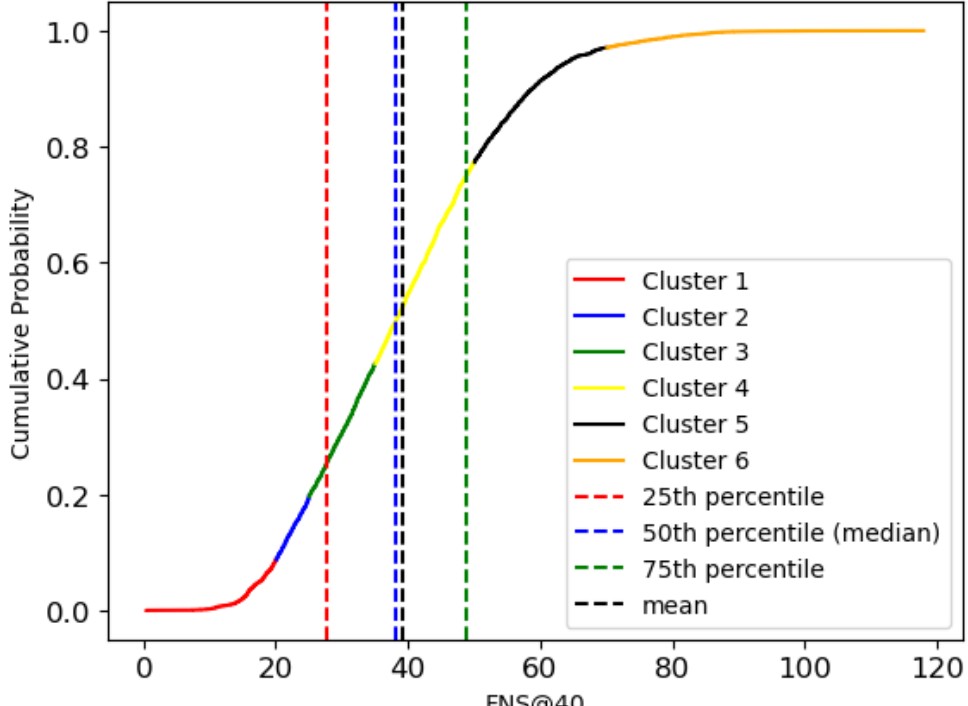

**Figure 8.** CFD of friction performance ratings and relatively statistical values.

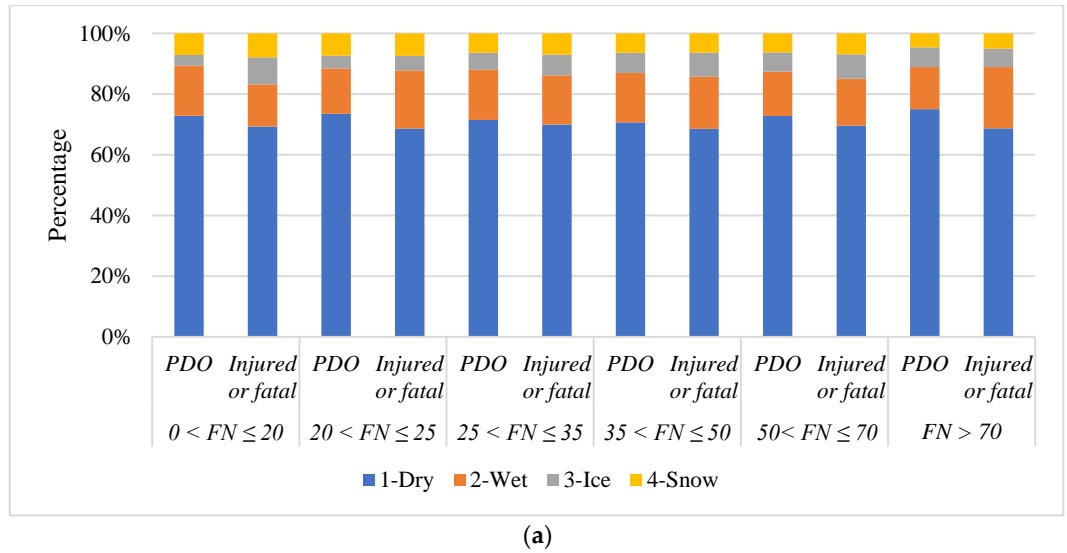

(**a**)

**Figure 9.** *Cont*.

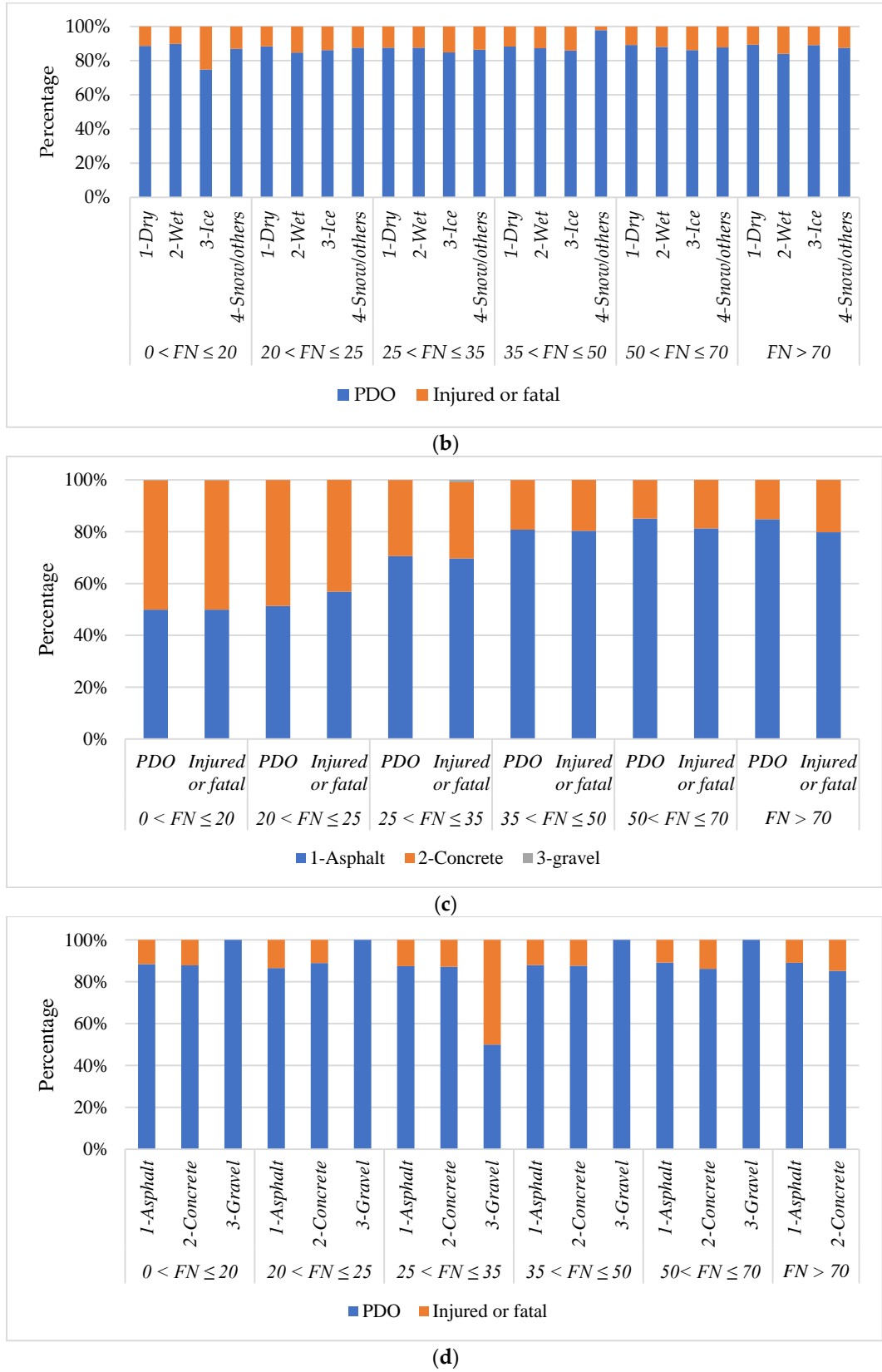

**Figure 9.** 100% stacked columns. (**a**) 100% stacked column of the surface condition under different crash-severity levels; (**b**) 100% stacked column of crash-severity level under different surface conditions; (**c**) 100% stacked column of surface material under different crash-severity levels; and (**d**) 100% stacked column of crash-severity level under different surface materials. Note: Gravel surfaces are not present when the friction number (FN) exceeds 70.

In addition to examining the associations between crash severity and the remaining three variables, additional factors such as vehicle speeds, following distances, and diverse crash causes also contribute valuable insights into the determination of pavement-performance ratings. Over 50% of crashes attributed to a short following distance occur within the FNS < 35 range, as shown in Figure 10a. Among these, within the range (20, 25], 66% of crashes are attributed to a small following distance. When FNS is less than or equal to 20, 64% of crashes are caused by this factor. Reduced skid resistance amplifies the challenges of braking, thereby increasing the likelihood of rear-end collisions. Approximately 95% of crashes resulting from a small following distance involve rear-end impacts, as shown in Figure 10b. When the FNS exceeds 35, there is a higher prevalence of crashes transpiring on interstates and US highways than on state roads. One notable characteristic of vehicles on interstates and US highways is their tendency to travel at higher speeds. The unsafe speeds can increase the risk of accidents. Based on Figure 10c, approximately 58% of crashes attributed to speed-related factors occur when the FNS is greater than 35. Furthermore, as depicted in Figure 10d, when the FNS is less than 35, multiple-vehicle crashes are more prevalent. However, as the FNS value gradually increases, the percentage of multiple-vehicle crashes decreases. This observation suggested that lower FNS values are associated with a higher likelihood of multiple-vehicle crashes, while higher FNS values are more closely linked to one-vehicle crashes. Therefore, in comparison to an FNS less than 35, the FNS larger than 35 exhibits a heightened susceptibility to collisions with animals or objects, as depicted in Figure 10b. Approximately 72% of such crashes occur within the FNS range. Particularly, within an FNS larger than 70, collisions with animals and other objects account for 43.68% of the total number of crashes of this nature.

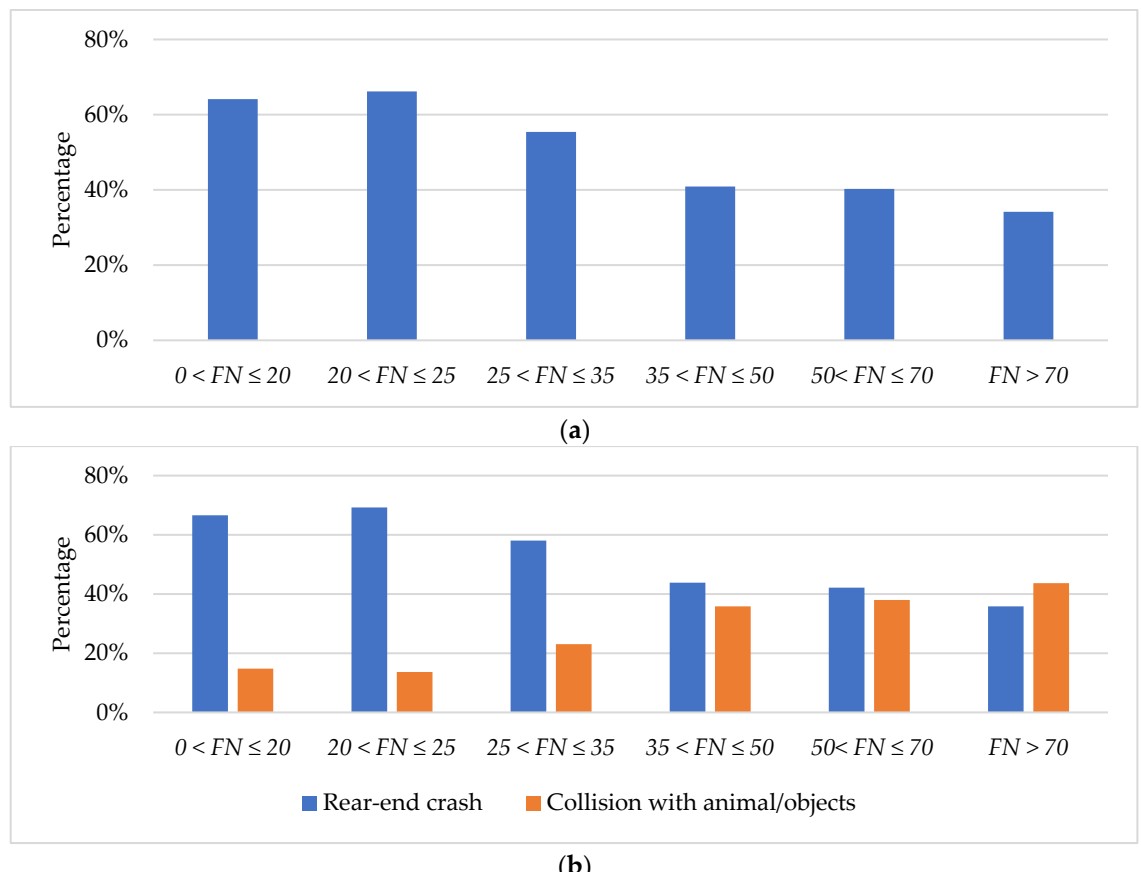

**Figure 10.** *Cont*.

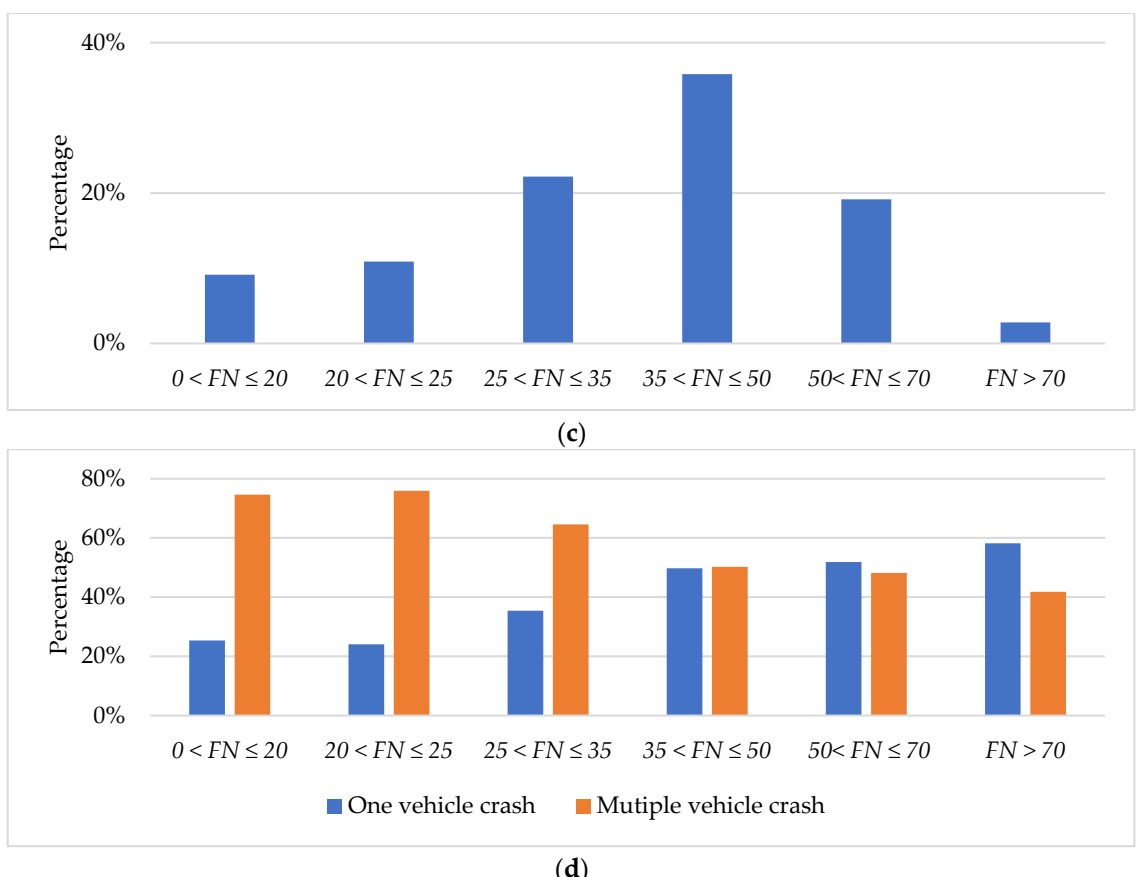

**Figure 10.** Cluster column chart. (**a**) crashes percentage caused by small following distance; (**b**) rear-end crash versus collisions with animals/objects; (**c**) distribution of crashes caused by unsafe speeds; and (**d**) one-vehicle crashes versus multiple-vehicle crashes.

Based on the above analysis, the characteristics of each friction performance rating can be summarized as follows:

- FNS $\in$ (0, 20]: Due to the extremely low FNS, the collision consequences in this range are much more serious than other FNS intervals, and extra attention should be paid to this type of road.

- FNS $\in$ (20, 25]: Similarly, the low FNS caused more serious crashes especially when the pavement surface is wet or covered with ice, snow, or other loose coverings. However, it is evident that shorter following distances are more likely to cause crashes. The underlying reason is attributed to inadequate friction, which hinders drivers from braking effectively and subsequently leads to rear-end collisions [15].

- FNS $\in$ (25, 35]: The 25th percentile of FNS falls within this range. There is a higher prevalence of gravel surfaces compared to other intervals. This increased presence of gravel surfaces contributes to a higher incidence of injuries and fatal accidents. Furthermore, an observable trend within this rating range is a shift from multiple-vehicle crashes to an increasing number of single-vehicle crashes. This shift suggests that the characteristics of the road surface within this FNS range may influence the nature and type of accidents that occur, emphasizing the importance of addressing road conditions, and promoting safe driving practices to mitigate the risk of accidents in this range.

- FNS $\in$ (35, 50]: The means and median of FNS were distributed within this range. Over 97% of crashes occurring on snow surfaces were classified as PDO crashes, with only 2.19% resulting in injury or fatal accidents. The proportion of PDO crashes and injured or fatal crashes on other surface conditions is almost the same as in other

ratings. Even though the majority of speed-related crashes occur within this range, it is believed that road surfaces with FNS In this range can ensure the safety of drivers traveling at required speed limitations.

- FNS $\in$ (50, 70]: The 75th percentile of FNS, which is around 50, is in close proximity to this range. Within this range, concrete roads exhibit a slightly higher susceptibility to injury and fatal crashes, while asphalt roads display a slightly higher likelihood of the occurrence of PDO crashes.
- FNS $\in$ (70, $\infty$): Similar to the case of FNS falling between 50 and 70, concrete roads with FNS values greater than 70 demonstrate an elevated vulnerability to injury and fatal crashes. Furthermore, collisions with animals/objects are more likely to happen in this range.

Provided below are the current practices of INDOT in managing pavement friction:

- A friction number of 20 is the flag value of friction that indicates necessary actions are warranted to restore pavement friction.
- A friction number ranging between 20 and 25 indicates the pavement friction may be lower than the flag value in the coming year(s), which can facilitate district pavement engineers to better plan pavement preservation, overlay, and resurfacing activities.
- A friction number of 35 is the minimum friction requirement for pavement warranty projects [43].
- A friction number greater than 70 is commonly required for new high friction surface treatment (HFST) that is commonly utilized at crash-prone areas with exceptionally high friction demand, such as sharp curves, ramps, bus stops, intersections, tunnel entrances, and steep grades [44].

Evidently, the ratings in Table 6 align effectively with the current practices. This may appear to be a coincidence, but it reflects the long-lasting and profound impact of INDOT's current practices in pavement-friction management on the overall pavement-friction performance of the road network.

In contrast to current practices that often provide arbitrary values lacking a strong scientific basis [2,15,30–33], this study employs a more comprehensive approach to friction performance ratings. The utilization of the DBSCAN-GMM algorithm and the Chi-square test effectively offers theoretical support for rating friction performance, going beyond the empirical determination of friction thresholds. Moreover, by simultaneously considering multiple background factors, this study establishes more robust ratings that accommodate the dynamic characteristics of road friction.

## 6. Conclusions

Pavement friction plays a critical role in ensuring road safety by preventing vehicle tires from sliding or skidding on the roadway pavement surface. To this end, this study analyzes the existing limitations of current methods utilized for assessing pavement-friction performance and developed DBSCAN-GMM models to establish pavement-friction performance ratings at a network level. By conducting an analysis and comparison of the results derived from one-, two-, and multi-dimensional DBSCAN-GMM models, and employing Chi-square tests to investigate the relationships between severity level and other variables within the multi-dimensional model, this study has successfully established six distinct categories for pavement-friction performance ratings. These categories are defined as (0, 20], (20, 25], (25, 35], (35, 50], (50, 70], and (70, $\infty$). These ratings effectively align with INDOT's current practice in managing pavement friction. Based on the findings presented in this paper, several key conclusions can be drawn as follows:

- In scenarios where the pavement exhibits an exceptionally low FNS (FNS < 25), the magnitude of collision consequences is markedly elevated in comparison to pavement characterized by higher FNS values, specifically when the road surface is subjected to the influence of water, ice, snow, or other unconsolidated substances. Additionally, in this range, there is an increasing likelihood of crashes attributable to small vehicles

following distances. This phenomenon can be attributed to the diminished frictional characteristics of the road surface, which hinder optimal braking capabilities and consequently contribute to an augmented incidence of rear-end collisions.

- The 25th percentile FNS falls within the range of 25 and 35. Within this range, there is a notable transition from multi-vehicle crashes to an elevated occurrence of single-vehicle crashes. Additionally, the presence of a higher number of gravel surfaces within this range leads to more severe crashes.
- The median and average FNS values fall within the range of 35 and 50. This range represents the first acceptable FNS range, characterized by a significantly low rate of injury or fatal crashes.
- For pavements with relatively high FNS (FNS > 50), concrete roads exhibit a slightly higher susceptibility to injury and fatal crashes compared to asphalt roads. Moreover, the increased likelihood of single-vehicle crashes within this range contributes to a higher probability of collisions with animals or objects.

These findings help better understand the relationship between pavement friction, crash severity, and other safety-related variables. They provide valuable insights for improving road safety and friction management. Furthermore, the comprehensive analysis and comparison with the outcomes obtained from the one-, two-, and multi-dimensional DBSCAN-GMM models offer compelling evidence to support the reliability and effectiveness of the DBSCAN-GMM hybrid clustering algorithm in accurately determining pavement-friction performance ratings.

**Author Contributions:** Conceptualization, S.L. and Y.J.; methodology, J.B., S.L., and Y.J.; software, J.B.; validation, J.B. and S.L.; formal analysis, J.B. and S.L.; investigation, J.B. and S.L.; resources, Y.J. and S.L.; data curation, J.B. and S.L.; writing—original draft preparation, J.B. and S.L.; writing—review and editing, Y.J. and S.L.; visualization, J.B.; supervision, Y.J. and S.L.; project administration, S.L.; funding acquisition, S.L. and Y.J. All authors have read and agreed to the published version of the manuscript.

**Funding:** This research was partially funded by the Joint Transportation Research Program (JTRP) between Purdue University and the Indiana Department of Transportation, grant number SPR-4646.

**Data Availability Statement:** Data and codes used in this paper are available upon request.

**Conflicts of Interest:** The authors declare no conflict of interest. The funders had no role in the design of the study; in the collection, analyses, or interpretation of data; in the writing of the manuscript; or in the decision to publish the results.

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
