# Peer review of "Determination of Safety-Oriented Pavement-Friction Performance Ratings at Network Level Using a Hybrid Clustering Algorithm"

_lubricants, doi:10.3390/lubricants11070275_

Round 1

Reviewer 1 Report

This paper investigated the determination of safety-oriented pavement friction performance ratings at the network level using a hybrid clustering algorithm. Here are some comments:

(1) The abstract should be improved. Please present the main results of the paper quantitatively.

(2) Why use "The Chi-square test"? How accurate is this method?

(3) On page 14, the lack of analysis for FNS ∈ (35, 50) and FNS > 70.

(4) The manuscript should be improved by adding critical discussions instead of briefly reporting the results of this study. This is an important issue and should be addressed effectively and thoroughly.

(5) In Table 5, why is the effect of surface material on FNS not significant when FNS > 70?

Reviewer 2 Report

  1.In Table 5. Chi-square test results, p-value is 0.1301 is accepted. What is the range of values accepted.How? 2.Table 6 is meaningless, where are the results?Kindly change it. 3.Scientific depth of explanation relevant to literature is missing for figure 8 a-d. 4.Include numerical results in the conclusion part. 5.In fig.8d, 25<fn<35 more injured or fatal happened. Justify the reason. 6.In fig.8a-d y axis label is missing. Include. 7. Table 4 is similar to table 6. Kindly remove it 8.What basis clustering was done on Fig.5, Specify the K value of clustering. 9. Specify the training and testing data sets. 10. What is the learning rate used to optimize the parameters? 11.Table 1 compares your proposed model with existing literature.  12. Abstract includes some numerical values of the proposed model. --
